# Investigating approach/avoidance tendencies in male AUD patients through a gait initiation task: An exploratory posturography study

Salvatore Campanella[1]*, Macha Dubuson[1], Maylis Pereira[1], Harold Mouras[2], Guillaume Leonard[3], Xavier Noel[1], Thierry Lelard[4]

1 Laboratoire de Psychologie Médicale et d'Addictologie, ULB Neuroscience Institute (UNI), CHU Brugmann-Université Libre de Bruxelles (U.L.B.), Belgium, 2 UR-UPJV: LNFP, Laboratoire de Neurosciences Fonctionnelles et Pathologies, Centre Universitaire de Recherche en Santé, rue René-Laennec, Amiens cedex, France, 3 Research Centre on Aging, CIUSSS de l'Estrie—CHUS, Sherbrooke, Canada, 4 UR-UPJV: APERE, Adaptations Physiologiques à l'Exercice et Réadaptation à l'Effort, UFR des Sciences du Sport, Université de Picardie-Jules-Verne, allée P. Grousset, Amiens, France

* salvatore.campanella@chu-brugmann.be or salvatore.campanella@ulb.be

## Abstract

### Introduction

This exploratory study investigates approach/avoidance tendencies in male patients with Alcohol Use Disorder (AUD) through a gait initiation task combined with posturography.

### Method

Seventy-four male participants (N = 74), including 47 AUD patients undergoing detoxification and 27 healthy controls, were exposed to alcohol-related, erotic, and neutral visual stimuli. Reaction times (RTs) and postural micromovements preceding gait initiation (forward/backward steps) were recorded to assess their predictive value for relapse (n = 13) or abstinence (n = 34) within two weeks post-detoxification.

### Results

A mixed ANOVA (2x2x3) revealed significant group differences in RTs to alcohol-related stimuli (Stimuli × Group interaction: p = .029), with relapsers showing slower responses to alcohol cues than abstainers for forward as well as backward steps. Additionally, postural micromovements before gait initiation (measured through the standard deviation of the center of pressure's position) showed a significant Stimuli × Group interaction (p = 0.05), with relapsers displaying increased micromovements when exposed to alcohol-related stimuli (p = .044).

**Data availability statement:** Research Ethics Committee imposed that data cannot be shared publicly because of patients' anonymity (as data contain potentially identifying or sensitive patient information). Data are available from the ULB Institutional Data Access for researchers who meet the criteria for access to confidential data. Only the names of the patients will be deleted and replaced by codes such as P1 (for patient 1), etc... Contact address is: Author: salvatore.campanella@ulb.be or through The Ethical committee that validated the study: christelle.demees@chu-brugmann.be All data have been collected by the authors.

**Funding:** S. Campanella and X Noel were funded by the Belgian Fund for Scientific Research (F.N.R.S., Belgium) and the Brugmann Foundation (UHC Brugmann, Brussels, Belgium). H Mouras would like to express his gratitude to "Institut National du Cancer" (INCA) and to "Institut pour la Recherche en Santé Publique" for their financial support of this project (SPAV1-23-009-2023-209).

**Competing interests:** The authors have declared that no competing interests exist.

## Conclusion

These findings suggest that relapsers exhibited distinct motor responses to alcohol-related stimuli, characterized by delayed RTs and increased postural instability. These early indicators of relapse risk highlight the potential of posturography as a clinical tool in alcohol detoxification programs.

## 1. Introduction

Despite positive outcomes associated with psychotherapy [1], social support [2] and anti-craving medication [3], the relapse rate, i.e., the moment in which a recently detoxified patient suffering from alcohol use disorder (AUD) loses control and starts again drinking, remains tremendously high: around 80% within 1 year of treatment [4]. This huge rate of relapses outlines the limitation of the current treatments proposed in clinical addiction units and the urgent need to find complementary and effective add-on tools [5].

The complexity of treating AUD certainly relates to the extensive array of contributing factors: social, educational, environmental, genetic, emotional, cognitive as well as brain-related factors can each trigger the onset and sustain the long-term maintenance of this addictive behavior [6]. A main paradox defining AUD refers to the explicitly stated desire to stop consumption (typically based on the awareness of huge deleterious familial, social, professional and health consequences) combined with the inability to refrain from alcohol drinking [7]. This "motivational conflict" between *intention* (motivation to stop consumption) and *behavior* (continued alcohol use and repeated relapses) represents a fundamental problem for clinicians working with AUD patients undergoing a detoxification program [8]. This discrepancy has been conceptualized through an "approach/appetitive versus an avoidance/aversive motivational system", which are independent but may be simultaneously activated, therefore leading to ambivalent motivational states [9]. At the neurocognitive level, two opposite systems were also described: an abnormal bottom-up (impulsive, limbic) system generating implicit attentional biases (increased attention specifically related to alcohol cues), a "wanting" (craving, i.e., desire to drink) behavior and an automatic approach tendency, which cannot be "regulated" by a weakened reflective/executive (control, prefrontal) system that cannot inhibit approach tendencies [10,11]. It has been proposed that a critical challenge in promoting abstinence among AUD patients is to reduce approach tendencies while enhancing avoidance ones [12,13]. Both tendencies were shown to be closely linked to AUD treatment outcome [14,15], and several studies revealed encouraging results, suggesting that training AUD patients to decrease approach tendencies or to increase avoidance ones significantly reduce craving and/or impact abstinence rate [16–26]. Nevertheless, it should be noted that (1) other studies involving risk groups also disclosed negative results, with the training approach showing no effect on craving scores and alcohol consumption when university students were tested [27,28]; and (2) *avoidance tendencies* were also shown to be associated in recently detoxified AUD patients with *a higher risk of*

*relapse* in the weeks following discharge from the rehabilitation program [29–31]. These heterogeneous and counterintuitive observations were certainly not compelling enough to question the efficacy of these training approaches, but further outlined the complexity of addressing AUD, as the relationship between approach/avoidance tendencies, attentional biases, craving, alcohol consumption and abstinence/relapse proves more intricate than expected. If the heterogeneity of the individual characteristics of the participants involved in these studies (AUD patients, risky populations such as students,…) can clearly influence the results, the method used to investigate approach-avoidance tendencies, using a Manikin or a Joystic task [29,32], which respectively implement a symbolic versus a real sensorimotor action, is also of critical importance. Accordingly, some studies have proposed the use of *posturography* to capture embodied "approach/avoidance" responses, aiming to capture more ecologically valid behavioral measures (i.e., actual bodily motion toward or away from an emotional cue), as the human postural control system has been shown to be coupled to neural structures that process affect [33].

Posturography is a well-suited tool to measure (unconscious) spontaneous whole-body movements in response to motivational targets, by recording displacements of the center of pressure (COP) when the participant stood up on a posturographic platform and was asked to stay still. The antero-posterior COP displacement (COP-AP) is typically assumed to index avoidance–approach processes: the mean COP-AP is used to score the extent to which a participant is leaning in the anterior or posterior direction during a trial. In this view, higher values represent greater forward (approach) lean and lower values represent backward (avoidance) lean [34]. As a matter of fact, unpleasant visual painful stimuli [35–37] and aversive (mutilation or polluted) scenes [38,39] evidenced withdrawal/avoidance freezing-type responses when compared to neutral stimuli, whereas preferred food pictures [40] and infant vocalizations inducing empathy to mothers [41] demonstrated significant postural movements of approaching when compared to neutral ones. Interestingly, Kavanagh [42] showed that college students confronted with scenes depicting social drinking displayed more COP-AP movements than those who saw neutral advertisements, indicating that alcohol advertisement influences postural control and self-reported craving levels. Also, Mirabella and colleagues [43] showed that angry and happy facial expressions altered forward gait initiation parameters differently, *only when relevant to ongoing goals* (angry faces increased RTs for forward gait initiation compared to happy ones). By contrast, when the valence of faces was task irrelevant, emotional expressions did not yield any effect. More recently, 51 AUD patients undergoing detoxification and 29 male controls were confronted to pictures depicting alcohol, sexual or neutral content [44]. Results showed that, if forward/backward micromovements while passively viewing alcohol or sexual content were not different between patients and controls, patients who relapsed several weeks following discharge from the rehabilitation program were more likely to lean back when viewing alcohol cues and to lean forward during exposure to sexual content than patients who remained abstinent. Such data indicated that posturography may be of help to evidence different patterns of action tendencies in SUD patients undergoing detoxification that may predict future abstinence or relapse.

Quiet standing to measure COP micromovements is therefore a common way to study approach/avoidance behaviors. Gait initiation involving a single step is another paradigm allowing to investigate directional effects of emotion [45], i.e., the ease with which a forward (approach) or a backward (avoidance) movement is organized and executed [46]. Indeed, when participants were asked to produce an incongruent forward step towards an unpleasant picture, elevated reaction times (RTs) were recorded, suggesting that participants had to override strong avoidance tendencies [47]. Further studies suggest that arousal more than valence and viewing duration affects forward gait initiation parameters [46]. More recently, a virtual reality study showed that initiation of a forward movement was facilitated in response to a (pleasant) flower and a backward movement in response to an (unpleasant) spider [48]. On this basis, the main objective of the present study is to ask AUD patients undergoing detoxification to initiate a forward or a backward step when confronted to alcohol-related, erotic (control rewarding cues) or neutral pictures, and to wonder whether step initiation data could predict future abstinence vs. relapse. In light of previous mixed findings suggesting that both approach and avoidance tendencies may be associated with relapse [21,29], the present study adopted an *exploratory* approach, with no predetermined hypotheses,

to investigate whether facilitation in forward or backward movement towards alcohol cues can predict abstinence or relapse. In any case, highlighting specific patterns of motor reactions which can help to predict future abstinence or relapse in recently detoxified AUD patients might be of the greatest relevance for clinicians.

## 2. Material and method

### 2.1. Participants

80 male participants took part in this study. Out of these 80 participants, 74 were included in the final analyses, with age ranging between 26 and 64 years (mean = 43.6 years ± 9). The 6 remaining participants were not analyzed, because one stopped the experiment before the end due to fatigue (n = 1) and because the others could not be reached at follow-up (n = 5). Among these final 74 participants, 47 were inpatients suffering from AUD (DSM-V criteria) (mean age = 43.4 years ± 9.3), recruited at the Brugmann University Hospital (Brussels, Belgium), and 27 were healthy controls. Patients were hospitalized for alcohol rehabilitation (four weeks detoxification program). The Brugmann Hospital detoxification program targets physical and psychological alcohol withdrawal. The patients were accommodated in double rooms and were not allowed to leave the premises during the first week. The patients received B complex vitamins and a fixed-dosage schedule of diazepam (a fixed-starting daily dose between 40–100 mg, depending on the severity of their alcohol dependence, for 5 consecutive days), followed by tapering to zero of 10 mg every two days [49]. The experiment took place at the end of the second or the beginning of the third week of hospitalization, ensuring that patients were no longer receiving diazepam. The inclusion criteria included heterosexual male French speakers between 18 and 65 years of age with severe AUD requiring alcohol rehabilitation, and desire to reduce their use. The exclusion criteria were personal history of neurological disorder, diagnosis of chronic psychotic disorder, use of alcohol or other illicit substance during the experiment. Homosexuality was an exclusion criteria, as the erotic pictures used in the study exclusively depicted heterosexual and female content. At follow-up, two weeks after discharge, 47 patients could be divided into 2 subgroups: a group of abstainers (n = 34, mean age = 43.11 years ± 9.65) and a group of relapsers (n = 13, mean age = 44.15 years ± 8.52). Finally, 27 healthy controls (mean age = 43.92 years ± 8.69) were enrolled. They were recruited by word of mouth and were compensated 30 euros for their participation. To be included as healthy subjects, participants should have no personal psychiatric/neurological history, not take medications, and be French-speaking. Participants with an Alcohol Use Disorder Identification Test (AUDIT) [50] cut-off score of >10 were excluded, as it has been showed to offer better specificity to tag AUD [51]. The Brugmann ethics committee approved our study (CE 2017/114). The experiment was conducted in the period between 2018 and 2020. All the participants provided their informed written consent in accordance with the declaration of Helsinki.

### 2.2. Tasks and procedure

#### 2.2.1. Step initiation tasks and procedure.
Participants were instructed to stand barefoot on a posturographic platform (Satel, Blagnac, France), positioned at a distance of 1 meter facing a 30-inch (76.2 cm) computer monitor. The monitor's tilt angle, adjustable between −45° and 45°, was set according to the participant's height to ensure optimal viewing and comfort. The screen had a resolution of 2560 x 1600 pixels.

Similarly to the protocol used in Noel's study [44], visual stimuli were presented in 2 conditions: alcohol (vs. neutral) and erotic (vs. neutral). Each condition was presented in a fixed order, with the alcohol condition always preceding the erotic condition. We are aware that this clearly involves an "order effect", but, as the alcohol-condition was the most interesting condition of our protocol, we would be sure to obtain these data in case of the patient would like to stop the experiment before the end. The alcohol condition comprised 10 alcohol-related images (e.g., people drinking alcohol) and 10 neutral images (e.g., landscapes or objects not containing alcohol). The erotic condition included 10 images of nude women or suggestive heterosexual couples and 10 neutral images, different from those used in the alcohol condition, but

still depicting landscapes or objects. These pictures were used as for the forward than for the backward step task. Therefore, each participant was confronted to a total of 80 trials. The alcohol-related images came from the Amsterdam Beverage Picture Set [52], while the neutral and erotic images came from the Nencki Affective Picture System [53]. E-Prime 3.0 was used to present the images and send content-specific triggers. The trials always proceeded in the same way, with a standardized presentation of the images. At the beginning of the trial, participants were asked to remain still on the platform with their arms at their sides. In a counterbalanced order, participants had to take either 1) a step forwards or 2) a step backwards, as quickly as possible as soon as the image disappeared. Each image was preceded by a fixation cross displayed for a random duration between 1,5 and 2,5 seconds. The fixation cross was used to focus the participant's attention on the center of the screen and provided a reference period for the position of the center of pressure. Then, the images were displayed for a random duration ranging from 2.5 to 4.5 seconds (see Fig 1). These random presentation times for the fixation cross and the stimuli were used to avoid habituation and frequency bias. When the image disappeared, the individual took a step forward or backward as quickly as possible (see Fig 2 for illustration). Once the posture had stabilized, the experimenter manually moved on to the next trial. Prior to beginning the task, participants were given three practice trials to ensure they understood the instructions. A brief pause of 2–3 minutes was provided between blocks.

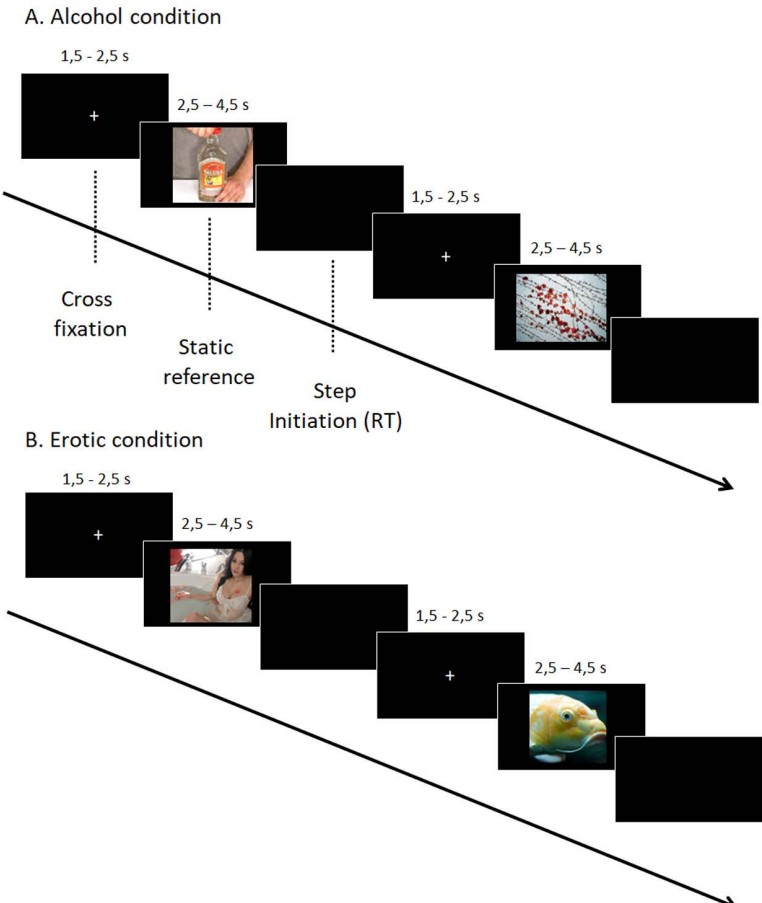

**Fig 1. Timeline of the experiment, including examples of neutral, alcohol-related, and erotic visual stimuli.**

**2.2.2. Questionnaires.** All participants (recruited from 01/01/2018–30/09/2020) were heterosexual males, and matched in age. Before starting the experiment, participants were administered a series of validated questionnaires to characterize the sample and assess potential confounding factors. We used the AUDIT to assess AUD severity (10 items, range 0–40); the Craving Experience Questionnaire (CEQ) [54], scoring the intensity (CEQ-Int) and the frequency (CEQ-Freq) of craving in the past week (22 items; range 22–153); the Beck Depression Inventory (BDI-II) [55] to index depressive symptoms (21 items, range 0–63); the Spielberger Trait Anxiety Inventory (STAI) [56], reporting the severity of anxious symptoms (20 items, range 20–80); the Impulsivity Behavior Scale (UPPS-P) [57], which assessed positive urgency (PU), negative urgency (NU), lack of premeditation (LPr), lack of perseverance (LPe), and sensation seeking (SS) (20 items, range 20–80); the Sexual Desires Inventory (SDI9) [58], measuring the dyadic (SDI-D) and the solitary (SDI-S) sexual desires (14 items, range 0–122); and a questionnaire about their alcohol consumption history (e.g., the age they lost control, the number of years of AUD, their participation in detoxification programs, and hallmarks (number of closed parents) of a family history of alcoholism). All details are summarized in Table 1.

**2.2.3. Follow-up data.** As recent studies showed early relapses [21,59], and in order to maximize the obtention of data, a follow-up procedure was planned for AUD patients *2 weeks* after the detoxification program. Relapse was defined as the reinstallation of a state of alcohol dependence and treated as a binary outcome variable [29,44]. Information was obtained by telephone, supplied directly by patients and confirmed by family members and the patient's regular doctor.

## 2.3. Data processing

The system's 3 strain gauges provided analog signals that were processed by the AcqKnowledge 4.2 software (Biopac Inc., Santa Barbara, CA) to determine the coordinates of the center of pressure. Variations in the position of the center of pressure (COP) were used to assess variations in posture linked to body movements. Anteroposterior displacement

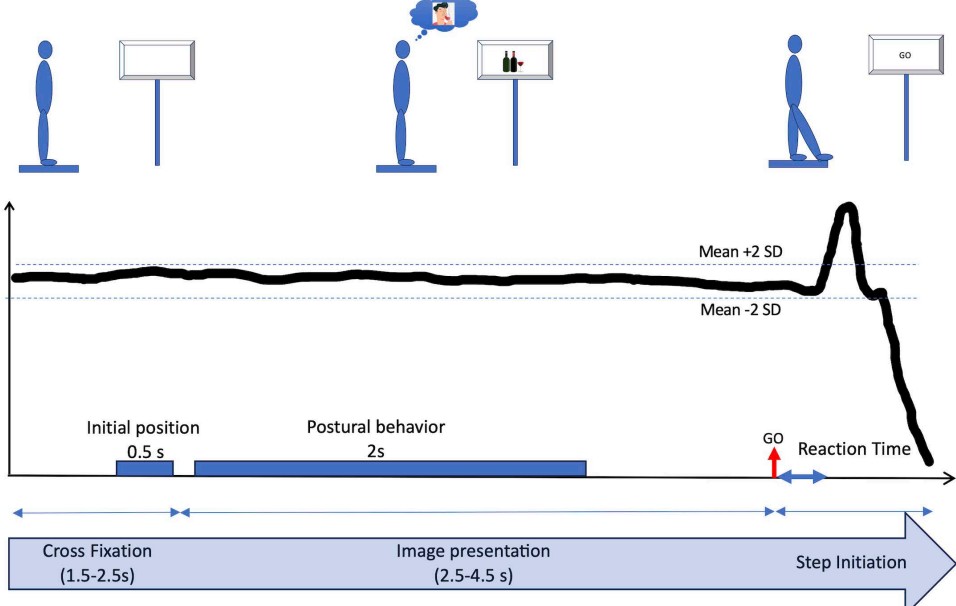

**Fig 2. During the first period, the participant had to remain still, maintaining a bipodal posture in front of the fixation cross.** The initial position of the center of pressure was calculated. During the second period, a visual stimulus was presented. To characterize the postural response, the standard deviation of the center of pressure position was calculated for a period of 2 seconds. The third period began with the disappearance of the image corresponding to the starting signal to initiate a step (GO). The reaction time was determined when the position of the center of pressure deviated by plus or minus 2 standard deviations from the mean position.

**Table 1. Mean (sd) of participants' characteristics for healthy controls, AUD patients, and subgroups of abstainers vs. relapsers after 2 weeks of discharge. Mean scores were compared thanks to independent t-tests between healthy controls and AUD patients as well as between abstainers and relapsers.**

| | Controls (N=27) Mean (sd) | Patients (N=47) Mean (sd) | Controls vs Patients (Student t-tests) | Abstainers (n=34) Mean (sd) | Relapsers (n=13) Mean (sd) | Abstainers vs Relapsers (Student t-tests) |
|---|---|---|---|---|---|---|
| Age | 43,9 (8,7) | 43,4 (9,3) | $t(72) = 0.238$, $p=.812$ | 43,1 (9,7) | 44,1 (8,5) | $t(45) = 0.339$; $p=.736$ |
| Education | 15,11 (2,25) | 13,4 (2,7) | $t(72) = 2.761$, $p=.007*$ | 13,5 (2,7) | 13,2 (2,7) | $t(45) = 0.428$; $p=.671$ |
| AUDIT | 5,4 (2,9) | 31,2 (5,8) | $t(72) = 21.68$, $p<.001**$ | 31 (5,5) | 31,9 (6,6) | $t(45) = 0.481$; $p=.629$ |
| CEQ-Intensity | / | 36,5 (22,3) | / | 39 (24,4) | 30,1 (14,9) | $t(45) = 1.231$; $p=.225$ |
| CEQ-Frequency | / | 24,6 (15) | / | 25,4 (16,6) | 22,7 (10,6) | $t(45) = 0.538$; $p=.593$ |
| BDI-II | 6,2 (5,1) | 17,4 (13,9) | $t(72) = 4.029$, $p<.001**$ | 15,8 (9,9) | 21,6 (21) | $t(45) = 1.284$; $p=.206$ |
| STAI-T | 50 (4,5) | 54,3 (6,2) | $t(72) = 3.177$; $p=.002*$ | 54,7 (6,2) | 53,6 (6,2) | $t(45) = 0.551$; $p=.584$ |
| UPPS – PU | 7,1 (2,7) | 7,6 (2,7) | $t(72) = 0.884$, $p=.380$ | 7,4 (2,8) | 8,4 (2,2) | $t(45) = 1.173$; $p=.247$ |
| UPPS – NU | 4,5 (3,4) | 6,4 (3,2) | $t(72) = 2.446$, $p=.017*$ | 6,8 (2,8) | 5,4 (4) | $t(45) = 1.412$; $p=.165$ |
| UPPS- LPr | 7,2 (2,6) | 8,3 (2,9) | $t(72) = 1.633$, $p=.107$ | 8,6 (2,9) | 7,4 (2,5) | UPPS- LPr $t(45) = 1.343$; $p=.186$ |
| UPPS – Lpe | 6,4 (1,9) | 8,8 (3,1) | $t(72) = 3.635$, $p=.001**$ | 8,9 (3,4) | 8,4 (2,2) | $t(45) = 0.566$; $p=.574$ |
| UPPS-SS | 6,4 (2,7) | 6,2 (2,9) | $t(72) = 0.190$, $p=.850$ | 5,9 (2,8) | 6,9 (3,4) | $t(45) = 0.988$; $p=.328$ |
| SDI-S | 18,1 (7,5) | 11,5 (7,6) | $t(72) = 3.634$, $p=.001**$ | 11,4 (7,8) | 11,8 (7,3) | $t(45) = 0.143$; $p=.887$ |
| SDI-D | 66,5 (15,7) | 52,7 (16,3) | $t(72) = 3.552$, $p=.001**$ | 54,7 (17,3) | 47,5 (12,6) | $t(45) = 1.351$; $p=.183$ |
| Age Loss control | / | 30,6 (9,4) | / | 30,5 (9,9) | 30,8 (8,4) | $t(45) = 0.112$; $p=.912$ |
| Years of AUD | / | 10,3 (9) | / | 10,5 (9,8) | 9,8 (6,8) | $t(45) = 0.246$; $p=.807$ |
| Nb of detox | / | 2,5 (3,2) | / | 2,4 (3,5) | 2,6 (2,6) | $t(45) = 0.191$; $p=.850$ |
| Family history | 0,4 (0,6) | 1,2 (1,3) | $t(72) = 2.983$; $p=.004*$ | 1,1 (1,4) | 1,3 (1,1) | $t(45) = 0.363$; $p=.719$ |

Abbreviation: Alcohol Use Disorder Inventory (AUDIT), Beck Depression Inventory (BDI), CEQ (Craving Experience Questionnaire), Impulsivity Behavior Scale (UPPS), urgency (PU), negative urgency (NU), lack of premeditation (LPr), lack of perseverance (LPe), sensation seeking (SS).

*Significant results at $p<.05$;

**Significant results at $p \leq .001$.

characteristics of the COP were analyzed with a custom MATLAB script. The initial position of the center of pressure corresponded, for each trial, to the mean position of the center of pressure during the last 500 ms of presentation of the fixation cross. First, we studied the impact of the emotional content of the image presentation in the preparatory phase of step

initiation. During the preparatory phase, oscillatory movements of the body were analyzed to verify whether the emotional content of the image modified the postural behavior and the attention paid to the disappearance of the visual stimulus. We calculated the standard deviation of the center of pressure's position as global indicator of body movement during the two seconds the image was presented. Then, participants were instructed to take a step (forward or backward) *as quickly as possible when the image disappeared.* To examine this voluntary response, a threshold was established for detecting the initiation of movement. Given physiological reaction times, any reaction time of less than 100 ms was considered an anticipatory movement and was excluded from the analysis [60,61]. On this basis, it is important to note the small ratio of rejected trials (mean of 2.4% for controls and of 4.2% for patients). Importantly, this ratio of rejected trials was also similar for future abstainers (4.3%) and relapsers (4.1%). The reaction time was determined from the moment when the position of the center of pressure deviated by more than two standard deviations from its initial position [62]. The analysis was conducted by two independent judges, each individually assessing the step (reaction time). In instances where there were substantial differences between their observations, we re-evaluated the data together and repeated the analyses to reach a consensus. This double analysis ensured that our findings were accurate and reliable.

## 2.4. Statistical analyses

Group comparisons were conducted between controls and AUD patients, as well as between abstinent and relapsed patients after 2 weeks of discharge. All participants were male and matched for age, as confirmed by an independent samples *t*-test (see Table 1 for all detailed statistics). As participants were confronted to separated blocks of neutral vs. alcohol-related and neutral vs. erotic-related stimuli, separated analyses were computed by blocks. Analyses of variance with the Greenhouse-Geisser correction applied where appropriate (ANOVA; 2 x 2 x 3 mixed factorial design) with 'Step' (forward, backward) and 'Stimuli' (neutral, alcohol) as the within-subject factors and 'Group' (controls; abstainers; relapsers) as the between-subject factor were computed. Post-hoc Bonferroni t-tests were used when appropriate.

## 3. Results

First, we compared RTs for gait initiation towards neutral vs. alcohol-related cues. Results showed a main effect of Step ($F(1,71) = 6.015$; $p = .017$; eta-squared = 0.078, observed power = 0.677), a main effect of Stimuli ($F(1, 71) = 43.177$; $p < .001$; eta-squared = 0.378, observed power = 1) and a *significant Stimuli x Group interaction* ($F(2, 71) = 3.731$; $p = .029$; eta-squared = 0.095, observed power = 0.665). In order to disentangle this interaction, mean RTs for forward and backward steps were computed for both neutral and alcohol stimuli, and were compared on each separate group using paired sample t-tests. Alcohol-related stimuli triggered slower motor reactions than neutral ones in all groups (Abstainers: 282 (50) vs. 297 (60) ms: $t(33) = 2.54$; $p = .016$; Relapsers: 269 (48) vs. 315 (57) ms: $t(12) = 5.23$; $p < .001$; Controls 278 (51) vs. 301 (53) ms: $t(26) = 3.654$; $p = .001$). Nevertheless, we also computed delta scores, i.e., difference RTs between stimuli (RTs Alcohol minus RTs Neutral), suggesting that larger delta indexed slower RTs to alcohol cues compared to neutral ones. A one-factor ANOVA showed a significant group effect ($F(2,73) = 3.731$; $p = .029$), with post-hoc Bonferroni t-tests disclosing *a significantlylarger delta RT for relapsers compared to abstainers when RTs for neutral trials were subtracted from RTs for alcohol pictures (mean difference of 30 ms; p = .024)*. No difference were found between abstainers and controls (mean difference of 8 ms; $p = 1$) as well as between relapsers and controls (mean difference of 21 ms; $p = .184$).

Second, a similar ANOVA 2 x 2 x3 with 'Step' (forward, backward) and 'Stimuli' (neutral, erotic) as the within-subject factors and 'Group' (controls; abstainers; relapsers) as the between-subject factor was computed. Here again we obtained main effect of Step ($F(1, 71) = 4.099$; $p = .047$; eta-squared = 0.055, observed power = 0.515) and Stimuli ($F(1, 71) = 5.922$; $p = .017$; eta-squared = 0.077, observed power = 0.670), suggesting that erotic stimuli and backward steps implied longer RTs. However, contrary to neutral vs. alcohol-related trials, any significant interaction Stimuli x Group emerged ($F(2, 71) = 0.35$; $p = .966$).

Third, as many previous papers showed that gait initiation could be affected by anticipatory postural adjustments [63,64], it was important to check whether our groups did differ on postural micromovements preceding forward and backward step initiation during the static phase. Body oscillations were characterized from the standard deviation of the antero-posterior position of the center of pressure to quantify postural variations. Again, two separated ANOVAs 2 x 2 x3 with 'Step' (forward, backward) and 'Stimuli' (neutral, alcohol or erotic) as the within-subject factors and 'Group' (controls; abstainers; relapsers) as the between-subject factor were computed. While any significant Group x Stimuli interaction was reported for the blocks with neutral-erotic stimuli ($F_{(2,68)} = 0.704$; $p = .498$), a significant Group x Stimuli interaction ($F_{(2,68)} = 3.193$; $p = .047$; eta-squared = 0.086, observed power = 0.592) was disclosed for neutral-alcohol pictures, suggesting through paired Student t-tests (comparing means of micromovements preceding both forward and backward steps) that only relapsers showed more postural changes ($t(12) = 2.306$; $p = .040$) prior gait initiation for alcohol pictures compared to neutral ones (mean of 7.05 mm vs. 5.27 mm), while no differences were observable for both controls ($t(25) = 0.109$; $p = .914$; 4.85 mm vs. 4.9 mm) and abstainers ($t(32) = 1.329$; $p = .193$; 6.74 mm vs. 5.91 mm) (see Table 2).

## 4. Discussion

In the present study, we used posturography to examine whether a forward and backward gait initiation task could provide insights into approach versus avoidance tendencies in AUD patients after detoxification therapy, with the aim of identifying variables that differentiate between those likely to relapse and those expected to remain abstinent 2 weeks after detoxification. Participants were presented with blocks of neutral and alcohol-related pictures, and asked to initiate a forward or a backward step. In order to confront participants with other rewarding stimuli, blocks with erotic pictures were also presented. Comparative measures were obtained in a group of healthy control participants. Interestingly, participants of all

**Table 2.** A. Mean (sd) of participants' RTs (ms) for forward and backward steps to neutral vs. alcohol stimuli. B. Mean (sd) of participants' RTs for forward and backward steps to neutral vs. erotic stimuli. C. Mean (sd) of the body center of pressure micromovement prior gait initiation in mm for forward and backward steps to neutral vs. alcohol stimuli. D. Mean (sd) of the body center of pressure micromovement prior gait initiation in mm for forward and backward steps to neutral vs. erotic stimuli.

|  |  | Controls | Relapsers | Abstainers |
|---|---|---|---|---|
| A | Forward – Alc | 291 (57) | 285 (50) | 294 (81) |
|  | Forward – Neu | 268 (59) | 254 (50) | 269 (73) |
|  | Backward – Alc | 303 (92) | 344 (93) | 309 (59) |
|  | Backward – Neu | 295 (82) | 285 (76) | 286 (59) |
|  |  |  |  |  |
| B | Forward – Ero | 280 (61) | 297 (51) | 278 (90) |
|  | Forward – Neu | 270 (55) | 288 (58) | 267 (73) |
|  | Backward – Ero | 311 (89) | 328 (87) | 288 (60) |
|  | Backward – Neu | 294 (81) | 307 (81) | 275 (63) |
|  |  |  |  |  |
| C | Forward – Alc | 5.9 (3.9) | 7.04 (3.8) | 4.9 (2.8) |
|  | Forward – Neu | 6.7 (4.6) | 5.27 (2.4) | 4.8 (2.1) |
|  | Backward – Alc | 5.9 (3.9) | 7.05 (3.9) | 4.9 (2.7) |
|  | Backward – Neu | 6.7 (4.7) | 5.27 (2.4) | 4.8 (2.1) |
|  |  |  |  |  |
| D | Forward – Ero | 5.7 (2.7) | 7.7 (5.3) | 5.1 (3.7) |
|  | Forward – Neu | 5.9 (3.2) | 6.9 (5.3) | 3.4 (1.3) |
|  | Backward – Ero | 5.1 (2.4) | 5.9 (3.1) | 4.3 (2.4) |
|  | Backward – Neu | 4.2 (1.3) | 4.9 (1.7) | 3.7 (1.6) |

groups were faster initiating a forward step towards neutral cues compared to alcohol-related ones, a consistent finding observed in abstinent AUD patients [65,66]. The present study revealed two main findings. First, relapsers exhibited longer RTs to alcohol-related pictures compared to neutral ones when initiating a forward or backward step, in contrast to abstainers. Moreover, only relapsers showed a different micromovement adjustment ratio during the static phase preceding gait initiation, when confronted to alcohol-related stimuli. For both RTs and spontaneous postural responses, no significant differences between future relapsers and abstainers were observed when erotic and neutral images were compared. This indicates a general and specific slowing in response to alcohol-related stimuli, regardless of movement direction.

The reported data suggest that, in general, backwards steps involved longer RTs than forward ones for all stimuli. Significant group differences suggest that, compared to abstainers, (1) relapsers demonstrate slowed down gait initiation RTs towards alcohol cues compared to neutral stimuli; and (2) an increased number of COP micromovements are observed in relapsers exposed to alcohol-related pictures. Taken together, these observations could underscore the influence of the patient's emotional state, as, for instance, a pleasant emotion generally facilitate the initiation of forward gait due to the approach-oriented salience of the movement [45]. In the present study, it appeared that our future relapsers disclosed more micromovements before gait initiation and higher RTs (as for backward than for forward steps) to alcohol trials only. These results may index an attentional bias triggered by alcohol cues that could be indicative of an unpleasant (or ambivalent) emotional state triggering conflictual motor-related reactions. Alcohol ambivalence, reflecting a motivational conflict tagging the co-occurrence, in an individual trying to quit or control his/her drinking behavior [67], of approach and avoidance inclinations toward drinking alcohol, has been shown to be a central feature of AUD [68]. Competing desires and their impact on craving are important features to be taken into account in the treatment of AUD [69]. Temporal analysis of COP displacements has previously been used to describe the complexity of posturographic responses toward motivational stimuli [70,71]. The increase in SD observed in the present study may reflect a combination of approach and avoidance behaviors that can only be investigated by temporal analysis over a longer period of time. To better discriminate and describe approach and avoidance phases, future studies should combine COP responses with a cinematic description of body movements [72].

The AUD patients included in the present study are patients undergoing a four-weeks alcohol detoxification program. However, it is well-known that emotional conflict resolution interfered with cognitive resources involved in motor planning [73]. Hence, the increased amplitude of early postural micromovements in response to alcohol trials may be linked to a transient defensive reaction, implemented during alcohol-related trials. Such observations are in line with previous reported data, suggesting that relapsers are more likely than abstainers to lean back when viewing alcohol cues [44], and that such avoidance tendencies predicted relapse at three months in recently detoxified alcoholic patients [29].

Even if we are aware that these reported differences, even if statistically significant, reflected subtle modifications (means of 30 ms and 1.78 mm), we would like to insist on the fact that, as in our previous posturography study of Noël and collaborators [44], these differences only emerged for alcohol-related pictures (not erotic ones) and allowed discriminating between future relapsers and abstainers. Of course, the current exploratory study suffers from some limitations, such as a small sample size (and a particularly small ratio of relapsers: n = 13), the fixed order effect (alcohol cues are always presented before erotic ones) and the focus on male patients that clearly reduce the generalizability of our results. However, these preliminary data stressed the utility to take into account even unconscious motor reactions in the context of alcohol use disorder, as such approach/avoidance motor behaviors may be totally different than self-reported desires or motivations, and sustain future relapse. Identifying such motor restrictions may be of the greatest clinical relevance, as neuromodulation tools, such as repetitive transcranial magnetic stimulation (rTMS), have already been shown to increase motor control and decrease craving in alcohol [74–76] and cannabis users [77]. A fascinating perspective could be therefore for further studies to combine these posturographic variables with the recording of brain activity [78], in order to index specific motor reactions associated with specific brain activity that may trigger relapse.

## Author contributions

**Conceptualization:** Salvatore Campanella, Macha Dubuson, Harold Mouras, Xavier Noel, Thierry Lelard.

**Data curation:** Macha Dubuson, Maylis Pereira.

**Formal analysis:** Macha Dubuson.

**Investigation:** Macha Dubuson.

**Methodology:** Thierry Lelard.

**Project administration:** Salvatore Campanella.

**Software:** Thierry Lelard.

**Supervision:** Salvatore Campanella, Thierry Lelard.

**Validation:** Xavier Noel, Thierry Lelard.

**Visualization:** Thierry Lelard.

**Writing – original draft:** Salvatore Campanella.

**Writing – review & editing:** Salvatore Campanella, Macha Dubuson, Harold Mouras, Guillaume Leonard, Xavier Noel, Thierry Lelard.

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
