## [Decision Letter · Decision Letter 0]

PONE-D-24-55094Investigating approach/avoidance tendencies in male AUD patients through a gait initiation task: An exploratory posturography studyPLOS ONE

Dear Dr. Campanella,

Thank you for submitting your manuscript to PLOS ONE. After careful consideration, we feel that it has merit but does not fully meet PLOS ONE’s publication criteria as it currently stands. Therefore, we invite you to submit a revised version of the manuscript that addresses the points raised during the review process.

 The paper was revised by two experts in the field. In general they found the paper interesting and the topic worth investigating, but they also raised several concerns that deserve to be carefully addressed by the authors. Comments about the selection of the analyzed population, statistical analysed, and the flow of the paper merit particular attention.

We look forward to receiving your revised manuscript.

Kind regards,

Alessandro Mengarelli

Academic Editor

PLOS ONE

Additional Editor Comments (if provided):

Reviewers' comments:

Reviewer's Responses to Questions

**Comments to the Author**

1. Is the manuscript technically sound, and do the data support the conclusions?

Reviewer #1: Yes

Reviewer #2: No

2. Has the statistical analysis been performed appropriately and rigorously? 

Reviewer #1: Yes

Reviewer #2: Yes

3. Have the authors made all data underlying the findings in their manuscript fully available?

Reviewer #1: No

Reviewer #2: No

4. Is the manuscript presented in an intelligible fashion and written in standard English?

Reviewer #1: Yes

Reviewer #2: Yes

5. Review Comments to the Author

Reviewer #1: The submission “Investigating approach/avoidance tendencies in male AUD patients through a gait initiation task: An exploratory posturography study” reports results from an innovative posturography approach/avoidance paradigm, tested in abstinent and relapsed patients with alcohol use disorder (AUD) and controls. Specifically, the authors investigate gait initiation in a step-forward and step-backwards task, as well as micromovements. Consistent with earlier reports, the micromovements during the task are associated with later relapse.

I generally support the interesting manuscript, which provides novel research data and contributes to the heterogenous observations regarding approach tendencies in AUT. Particularly, their innovative method provides novel insights into the behavioral underpinnings of alcohol use disorder and its relapse. Below, I noted a number of issues that could provide to a more comprehensive reporting of the results.

COMMENTS

- Their abstract should include the information that the patient group included abstinence and relapse, and also the sub-group sizes should be stated.

- Abstract: In which direction were responses slower (both?), and which micromovements were increased? This should be more explicitly spelled out in abstract, as well as in the results section. Please consider if it can be stated whether an alcohol avoidance bias was present in all groups, given that all participants were faster initiating forward movements for neutral stimuli. This seems to be consistent with the approach/avoidance and implicit measures literature in AUD patients in abstinence, but not risk groups, e.g.

Dandaba, M., Serra, W., Harika-Germaneau, G., Silvain, C., Langbour, N., Solinas, M., ... & Chatard, A. (2020). Predicting relapse in patients with severe alcohol use disorder: The role of alcohol insight and implicit alcohol associations. Addictive behaviors, 107, 106433.

Schwippel, T., Schroeder, P. A., Hasan, A., & Plewnia, C. (2022). Implicit measures of alcohol approach and drinking identity in alcohol use disorder: A preregistered double‐blind randomized trial with cathodal transcranial direct current stimulation (tDCS). Addiction Biology, 27(4), e13180.

- Introduction: I suggest a critical revision of the section on approach-avoidance trainings. The authors argue and cite that “the training approach showing no effect on craving scores and alcohol consumption”. I do not agree that these findings (from risk groups) are compelling enough to question the efficacy of AAT on relapse prevention, cited earlier, recommended in clinical guidelines in several countries. Also, I do not think that this is required as a primary motivation of the study. The mechanism of the AAT is not clear – and it might not even be related to approach/avoidance tendencies. However, although the AAT as a training works fine, its diagnostic properties are poor (e.g. reliability, sensitivity). Thus, it would be sufficient to discuss the unclear prediction from assessments on AAT, and that those assessments could be improved (better precision, better reliability, higher objectivity) by using spatial parameters in the posturography / gait initiation task.

- When discussing the studies in detail, it would be beneficial to highlight whether they tested AUD patients in detox, or risk-groups. Studies with AUD patients in the clinic have often reported implicit avoidance tendencies.

- Is ambivalence towards alcohol possibly a better predictor than approach or avoidance? And is ambivalence reflected by the COP recordings?

- Several methodological details, that might play into the reported findings, could be included: 1) When during the 4-wk detoxification were participants tested, and was time of testing identical for relapse and abstaining patients; 2) Were all tested males heterosexual? Given the stimuli used in the control task, this could be relevant; I assume it is part of the SDI9 and should be mentioned 3) How many trials / % of data were removed due to anticipatory responses (p. 14)?

- Were the stimuli matched? Figure 4 suggests that there might exist several differences in stimulus composition, complexity, colors and so on; such basic features can influence approach and avoidance movements.

- It is very unfortunate that the order of conditions was fixed and not counterbalanced. We can expect practice effects and lower variability in the second assessments. The authors were worried that patients would like to stop the experiment; how often was the experiment stopped? Due to the fixed order, it can’t be concluded that the observed findings are specific to alcohol cues. This should be made explicit in the discussion.

- What is the “D” test statistic – or is it a typo and it should refer to “F”-values obtained from ANOVA?

- Critical and non-significant statistical results should be reported, as well (e.g., for the two-way interaction on p. 16).

- Discussion: I missed that “gait initiation velocity” (p. 18) was measured or reported and suggest the authors use response times, unless they also performed kinematic analysis; also, the speculations on “unpleasant emotional states” would be very easy to assess. The discussion should focus more concise on the reported results.

- At the same time, there is too little discussion on the direction of behavior. Table 2 at least suggests that the shorter RT for relapsers seems at least partially driven by a longer RT when stepping backwards.

- Figure 2: A continuous timeline could be introduced to help map the shown events. Also, please check if the event length corresponds to the trial duration/s.

Typo

- P. 16, 7.05 mm vs. 5.27 mm

Reviewer #2: In this study, the authors investigated approach-avoidance tendencies toward alcohol-related stimuli in a group of patients with alcohol use disorder and a group of healthy controls. The topic is relevant, and the decision to study approach-avoidance behaviors through postural movements is an interesting approach that provides different insights compared to traditional computer-based approach-avoidance tasks. However, I have some comments primarily concerning methodological aspects, as well as the analysis and interpretation of the data.

1. All participants are male; however, the authors did not report this as an inclusion criterion in the participant section. Additionally, considering that the erotic condition included images of nude women, for this to have ecological validity, the male participants should also have been heterosexual. I wonder if this was controlled for and included as an inclusion criterion. If this was not accounted for, I suggest that the authors include it as a limitation of the study.

2. In the inclusion/exclusion criteria for the control group, there is no mention of alcohol consumption. The controls complete the AUDIT, and I wonder if the authors have considered a cutoff score beyond which a participant would be excluded due to a high score. If this is the case, I ask the authors to include this cutoff among the exclusion criteria.

3. I find the organization of the methodological section somewhat confusing and difficult to follow. I suggest that the authors reorganize this part. For example, they could include a "Procedure" section where they incorporate the first four lines of the "Questionnaires" section, then explain that all participants had to complete two conditions in a fixed order (lines 224–231, page 11), and briefly introduce the participants' task (e.g. line 255- 259 page 12) before providing a more detailed explanation in a subsequent section. Moreover, in the visual stimuli section, I would include only the description of the images (line 231-238), while the description of their presentation and the trials should be integrated into the task description in the following section.

4. The description of the experimental task provided on page 12 is not particularly clear. The statement that the experiment was divided into two tasks gives the impression that the two tasks were performed sequentially, one after the other, whereas they were actually two different phases of the same trial. I find the description provided in the caption of Figure 2 much clearer; the authors could adopt that description in the main text as well.

5. I ask the authors to modify Table 1 to APA format and include it in the results section, after specifying in a statistical analysis section which test was used to compare the groups.

6. My main concern regards the interpretability of the results obtained. For several reasons, I fear that the validity of the analyses may be compromised. First, the sample sizes are very different. There are only 13 relapsers compared to 34 abstainers and 27 healthy controls. In addition to the sample size discrepancy, almost all the results discussed refer to the relapser group, which consisted of only 13 participants. I acknowledge that this is not the authors' "fault," but they should acknowledge this in the limitations section. Furthermore, if I understand correctly, for each condition included in the analyses (e.g., forward step for alcohol stimuli), each participant completed only 5 trials, which is few considering the limited number of participants.

7. The statistical analyses (e.g., lines 312-317, lines 334-336…) should be included in a dedicated section within the methods.

8. The description of the results of the post-hoc one-way ANOVA on the delta score is very difficult to understand, and I don't think it is entirely correct as it is written. If I understand correctly, a larger delta means that the movements in response to alcohol stimuli are slower compared to those in response to neutral stimuli. This effect is present in all groups, but with a difference between relapsers and abstainers. Reading the text, it seems that relapsers are faster than abstainers to respond to alcohol, suggesting a smaller delta. However, looking at the means (line 325), it appears that abstainers have a smaller delta (15) and relapsers have a much larger one (48). This result would suggest that relapsers are even more slowed down by alcohol stimuli, which is not what is stated. I also cannot understand what the numbers in parentheses refer to (e.g., 30 ms). I ask the authors to clarify this point, as it is one of the main analyses of their paper.

9. Once point 8 is clarified, I will be able to better assess the discussion. If the interpretation I provided above is correct, I have some doubts regarding how this result has been discussed. The authors state that relapsers demonstrate reduced gait initiation toward alcohol cues. However, the analyses show an effect that is not dependent on the direction of the step. So, from what I understand from the results, relapsers are slower to initiate gait in the presence of alcohol stimuli, but both in approach and avoidance. In fact, looking at the means, relapsers seem to be particularly slow at moving away from alcohol (344 ms) rather than approaching it (285 ms). These results are therefore not as contradictory to the authors' hypotheses, and there is no need to discuss a negative emotional state.

6. PLOS authors have the option to publish the peer review history of their article (what does this mean? ). If published, this will include your full peer review and any attached files.

**Do you want your identity to be public for this peer review?** For information about this choice, including consent withdrawal, please see our Privacy Policy .

Reviewer #1: **Yes: ** Philipp Schroeder

Reviewer #2: **Yes: ** Valentina Meregalli

---

## [Author Response · Author response to Decision Letter 1]

23 Apr 2025

We thank the Editor and both reviewers for their thorough job which helped us a lot to increase the quality of our paper. We hope you will be satisfied by our modifications, and we remain at your full disposal for any further requirement. Modifications can be checked at pages reported below in the MS with tracked changes (Revision1Edited.doc).

Reviewer #1: The submission “Investigating approach/avoidance tendencies in male AUD patients through a gait initiation task: An exploratory posturography study” reports results from an innovative posturography approach/avoidance paradigm, tested in abstinent and relapsed patients with alcohol use disorder (AUD) and controls. Specifically, the authors investigate gait initiation in a step-forward and step-backwards task, as well as micromovements. Consistent with earlier reports, the micromovements during the task are associated with later relapse.

I generally support the interesting manuscript, which provides novel research data and contributes to the heterogenous observations regarding approach tendencies in AUT. Particularly, their innovative method provides novel insights into the behavioral underpinnings of alcohol use disorder and its relapse. Below, I noted a number of issues that could provide to a more comprehensive reporting of the results.

We thank Reviewer 1 for his/her comments.

COMMENTS

- Their abstract should include the information that the patient group included abstinence and relapse, and also the sub-group sizes should be stated.

This has been included in the abstract.

- Abstract: In which direction were responses slower (both?), and which micromovements were increased? This should be more explicitly spelled out in abstract, as well as in the results section. Please consider if it can be stated whether an alcohol avoidance bias was present in all groups, given that all participants were faster initiating forward movements for neutral stimuli. This seems to be consistent with the approach/avoidance and implicit measures literature in AUD patients in abstinence, but not risk groups, e.g.Dandaba, M., Serra, W., Harika-Germaneau, G., Silvain, C., Langbour, N., Solinas, M., ... & Chatard, A. (2020). Predicting relapse in patients with severe alcohol use disorder: The role of alcohol insight and implicit alcohol associations. Addictive behaviors, 107, 106433.

Schwippel, T., Schroeder, P. A., Hasan, A., & Plewnia, C. (2022). Implicit measures of alcohol approach and drinking identity in alcohol use disorder: A preregistered double‐blind randomized trial with cathodal transcranial direct current stimulation (tDCS). Addiction Biology, 27(4), e13180.

This has been included in the abstract. We also thank the reviewer for the suggestion and for the references which have been included in the discussion (p 23) and reference sections (p 31).

- Introduction: I suggest a critical revision of the section on approach-avoidance trainings. The authors argue and cite that “the training approach showing no effect on craving scores and alcohol consumption”. I do not agree that these findings (from risk groups) are compelling enough to question the efficacy of AAT on relapse prevention, cited earlier, recommended in clinical guidelines in several countries. Also, I do not think that this is required as a primary motivation of the study. The mechanism of the AAT is not clear – and it might not even be related to approach/avoidance tendencies. However, although the AAT as a training works fine, its diagnostic properties are poor (e.g. reliability, sensitivity). Thus, it would be sufficient to discuss the unclear prediction from assessments on AAT, and that those assessments could be improved (better precision, better reliability, higher objectivity) by using spatial parameters in the posturography / gait initiation task.

We totally agree with the reviewer. We do not intend to minimize the impact of AAT in relapse prevention, but just to outline that some negative results have also been reported. We clarified this point at page 4.

- When discussing the studies in detail, it would be beneficial to highlight whether they tested AUD patients in detox, or risk-groups. Studies with AUD patients in the clinic have often reported implicit avoidance tendencies.

This has been done where appropriate (pp 4, 5, 23).

- Is ambivalence towards alcohol possibly a better predictor than approach or avoidance? And is ambivalence reflected by the COP recordings?

We thank the reviewer for this interesting comment, which was added in the Discussion section (p 24).

- Several methodological details, that might play into the reported findings, could be included: 1) When during the 4-wk detoxification were participants tested, and was time of testing identical for relapse and abstaining patients; 2) Were all tested males heterosexual? Given the stimuli used in the control task, this could be relevant; I assume it is part of the SDI9 and should be mentioned 3) How many trials / % of data were removed due to anticipatory responses (p. 14)?

1) This was clarified at page 7; 2) Yes, this was also clarified in the text at pages 7-8; and 3) Each participant was confronted to a total of 80 trials. Therefore, we obtained, for the 27 controls, a total of 2160 trials, and for the 47 patients, a total of 3760 trials. We computed the number of rejected trials due to anticipatory responses (<100ms), and we observed a small ratio of rejected trials, even if this ratio was a bit more important for patients (161 rejected trials on a total of 3760 = 4.2%) than for controls (51 trials on a total of 2160 = 2.4%). Importantly, the number of rejected trials was similar for future abstainers (n=34; 118 trials on a total of 2720 = 4.3%) and relapsers (n=13; 43 trials on a total of 1040 = 4.1%), so that reported differences between these two groups could not be attributed to the amount of rejected (anticipatory) RTs. This was reported on page 18.

- Were the stimuli matched? Figure 4 suggests that there might exist several differences in stimulus composition, complexity, colors and so on; such basic features can influence approach and avoidance movements.

We agree with the Reviewer that the presented pictures present some physical differences that may of course impact approach and/or avoidance tendencies towards neutral and alcohol/erotic pictures. However, we would like to stress that (1) these pictures were taken from validated picture batteries (the Amsterdam Beverage Picture Set [52], and the Nencki Affective Picture System [53]); and (2) the main goal of the present study was to compare our 3 groups (controls, abstinent, relapsers) who were then confronted with the same pictures, so that the reported differences between stimuli between groups should not be subtended by these parameters.

- It is very unfortunate that the order of conditions was fixed and not counterbalanced. We can expect practice effects and lower variability in the second assessments. The authors were worried that patients would like to stop the experiment; how often was the experiment stopped? Due to the fixed order, it can’t be concluded that the observed findings are specific to alcohol cues. This should be made explicit in the discussion.

We also totally agree. This was mentioned as a clear limitation (mainly regarding potential differences between alcohol and erotic pictures), even if here too it seems to us important to outline that only this procedure ensured that future abstinent and relapsers were confronted to the same order of stimuli presentation (so that group differences may then not be due to an order effect). Only one patient stopped the experiment due to fatigue, this was mentioned in the text at page 7.

- What is the “D” test statistic – or is it a typo and it should refer to “F”-values obtained from ANOVA?

Thank you for this remark. This was changed in the text where appropriate.

- Critical and non-significant statistical results should be reported, as well (e.g., for the two-way interaction on p. 16).

This was added at pages 20-21.

- Discussion: I missed that “gait initiation velocity” (p. 18) was measured or reported and suggest the authors use response times, unless they also performed kinematic analysis; also, the speculations on “unpleasant emotional states” would be very easy to assess. The discussion should focus more concise on the reported results.

Thank you for noting this: we just mean gait initiation RTs. This was clarified in the text at page 23.

- At the same time, there is too little discussion on the direction of behavior. Table 2 at least suggests that the shorter RT for relapsers seems at least partially driven by a longer RT when stepping backwards.

We agree with the Reviewer that longer RTs were observed for backward steps. However, It must be reminded that the triple step x stimuli x group interaction was not significant (F(2, 71)= 1.498; p=.230). Therefore, we can hardly discuss the reported significant interaction group x stimuli by including the step factor. However, we clearly highlight that backwards steps involved longer RTs than forward ones on page 23.

- Figure 2: A continuous timeline could be introduced to help map the shown events. Also, please check if the event length corresponds to the trial duration/s.

Figure 2 has been updated.

Typo

- P. 16, 7.05 mm vs. 5.27 mm

This was corrected.

Thank you.

Reviewer #2: In this study, the authors investigated approach-avoidance tendencies toward alcohol-related stimuli in a group of patients with alcohol use disorder and a group of healthy controls. The topic is relevant, and the decision to study approach-avoidance behaviors through postural movements is an interesting approach that provides different insights compared to traditional computer-based approach-avoidance tasks. However, I have some comments primarily concerning methodological aspects, as well as the analysis and interpretation of the data.

We thank Reviewer 2 for his/her comments.

1. All participants are male; however, the authors did not report this as an inclusion criterion in the participant section. Additionally, considering that the erotic condition included images of nude women, for this to have ecological validity, the male participants should also have been heterosexual. I wonder if this was controlled for and included as an inclusion criterion. If this was not accounted for, I suggest that the authors include it as a limitation of the study.

We clarified this important point at pages 7-8.

2. In the inclusion/exclusion criteria for the control group, there is no mention of alcohol consumption. The controls complete the AUDIT, and I wonder if the authors have considered a cutoff score beyond which a participant would be excluded due to a high score. If this is the case, I ask the authors to include this cutoff among the exclusion criteria.

We thank the reviewer for this important remark. This was added in the text at page 8, and the appropriate reference (used to choose the cut-off score; Babor et al., 2001) was added in the reference section (p 30).

3. I find the organization of the methodological section somewhat confusing and difficult to follow. I suggest that the authors reorganize this part. For example, they could include a "Procedure" section where they incorporate the first four lines of the "Questionnaires" section, then explain that all participants had to complete two conditions in a fixed order (lines 224–231, page 11), and briefly introduce the participants' task (e.g. line 255- 259 page 12) before providing a more detailed explanation in a subsequent section. Moreover, in the visual stimuli section, I would include only the description of the images (line 231-238), while the description of their presentation and the trials should be integrated into the task description in the following section.

We completely reorganized the methodological section. We hope it will be clearer now.

4. The description of the experimental task provided on page 12 is not particularly clear. The statement that the experiment was divided into two tasks gives the impression that the two tasks were performed sequentially, one after the other, whereas they were actually two different phases of the same trial. I find the description provided in the caption of Figure 2 much clearer; the authors could adopt that description in the main text as well.

This was also clarified in the text. Figure 2 was also updated.

5. I ask the authors to modify Table 1 to APA format and include it in the results section, after specifying in a statistical analysis section which test was used to compare the groups.

This was done.

6. My main concern regards the interpretability of the results obtained. For several reasons, I fear that the validity of the analyses may be compromised. First, the sample sizes are very different. There are only 13 relapsers compared to 34 abstainers and 27 healthy controls. In addition to the sample size discrepancy, almost all the results discussed refer to the relapser group, which consisted of only 13 participants. I acknowledge that this is not the authors' "fault," but they should acknowledge this in the limitations section. Furthermore, if I understand correctly, for each condition included in the analyses (e.g., forward step for alcohol stimuli), each participant completed only 5 trials, which is few considering the limited number of participants.

We totally agree and this was clearly mentioned as a main limitation of our study. We also clarified in the text that participants were confronted to 10 trials of each condition. This was done on pages 9 and 25.

7. The statistical analyses (e.g., lines 312-317, lines 334-336…) should be included in a dedicated section within the methods.

This was done, thank you (p 19).

8. The description of the results of the post-hoc one-way ANOVA on the delta score is very difficult to understand, and I don't think it is entirely correct as it is written. If I understand correctly, a larger delta means that the movements in response to alcohol stimuli are slower compared to those in response to neutral stimuli. This effect is present in all groups, but with a difference between relapsers and abstainers. Reading the text, it seems that relapsers are faster than abstainers to respond to alcohol, suggesting a smaller delta. However, looking at the means (line 325), it appears that abstainers have a smaller delta (15) and relapsers have a much larger one (48). This result would suggest that relapsers are even more slowed down by alcohol stimuli, which is not what is stated. I also cannot understand what the numbers in parentheses refer to (e.g., 30 ms). I ask the authors to clarify this point, as it is one of the main analyses of their paper.

We agree with the reviewer and we tried to clarify this point. This could also been due to the fact that an error was present in the results section, as of course the delta RTs was LARGER (and not shorter as previously mentioned) for relapsers to alcohol trials. This was clarified at page 20.

9. Once point 8 is clarified, I will be able to better assess the discussion. If the interpretation I provided above is correct, I have some doubts regarding how this result has been discussed. The authors state that relapsers demonstrate reduced gait initiation toward alcohol cues. However, the analyses show an effect that is not dependent on the direction of the step. So, from what I understand from the results, relapsers are slower to initiate gait in the presence of alcohol stimuli, but both in approach and avoidance. In fact, looking at the means, relapsers seem to be particularly slow at moving away from alcohol (344 ms) rather than approaching it (285 ms). These results are therefore not as contradictory to the authors' hypotheses, and there is no need to discuss a negative emotional state.

We thank the reviewer for this remark. As also mentioned in response to Reviewer 1, the triple interaction step x stimuli x group was not significant (F(2, 71)= 1.498; p=.230), so that between-group differences between forward and backward steps can hardly be discussed. However, an important result is also that only relapsers showed more micromovements before gait initiation to alcohol cues. This could be an index of

---

## [Decision Letter · Decision Letter 1]

PONE-D-24-55094R1Investigating approach/avoidance tendencies in male AUD patients through a gait initiation task: An exploratory posturography studyPLOS ONE

Dear Dr. Campanella,

Thank you for submitting your manuscript to PLOS ONE. After careful consideration, we feel that it has merit but does not fully meet PLOS ONE’s publication criteria as it currently stands. Therefore, we invite you to submit a revised version of the manuscript that addresses the points raised during the review process.

 Reviewers provided a good feedback about the revised paper, recognizing a significant improvement. However, one reviewer raised an additional comment that requires attention. I would encourage the authors to address also this final observation in order to improve the discussion of the results, and the overall quality of the paper.

We look forward to receiving your revised manuscript.

Kind regards,

Alessandro Mengarelli

Academic Editor

PLOS ONE

Journal Requirements:

Reviewers' comments:

Reviewer's Responses to Questions

**Comments to the Author**

1. If the authors have adequately addressed your comments raised in a previous round of review and you feel that this manuscript is now acceptable for publication, you may indicate that here to bypass the “Comments to the Author” section, enter your conflict of interest statement in the “Confidential to Editor” section, and submit your "Accept" recommendation.

Reviewer #1: All comments have been addressed

Reviewer #2: All comments have been addressed

2. Is the manuscript technically sound, and do the data support the conclusions?

Reviewer #1: Yes

Reviewer #2: Partly

3. Has the statistical analysis been performed appropriately and rigorously? 

Reviewer #1: Yes

Reviewer #2: Yes

4. Have the authors made all data underlying the findings in their manuscript fully available?

Reviewer #1: No

Reviewer #2: No

5. Is the manuscript presented in an intelligible fashion and written in standard English?

Reviewer #1: Yes

Reviewer #2: Yes

6. Review Comments to the Author

Reviewer #1: The authors have addressed my comments and provided a corrected revision. I have no further objections. Thank you for sharing your interesting findings!

Reviewer #2: The authors have thoroughly revised the manuscript, which is now much clearer, more methodologically rigorous, and easier to follow.

I have only one final comment for the authors, which concerns one of the early points in the Discussion section. Specifically, the authors state that "participants of all groups were faster initiating a forward movement... suggesting an alcohol avoidance bias." However, the results show a main effect of stimulus type, but not an interaction with movement direction. Therefore, I do not believe it is appropriate to interpret this as evidence of an avoidance bias. Rather, the data seem to indicate a general slowing in response to alcohol-related stimuli, regardless of movement direction.

Apart from this, in my view the paper is now ready for publication.

7. PLOS authors have the option to publish the peer review history of their article (what does this mean? ). If published, this will include your full peer review and any attached files.

**Do you want your identity to be public for this peer review?** For information about this choice, including consent withdrawal, please see our Privacy Policy .

Reviewer #1: **Yes: ** Philipp A. Schroeder

Reviewer #2: **Yes: ** Valentina Meregalli

---

## [Author Response · Author response to Decision Letter 2]

6 Jun 2025

We thank the Editor and both reviewers for their thorough job which helped us a lot to increase the quality of our paper. We hope you will be satisfied by our modifications, and we remain at your full disposal for any further requirement. Modifications can be checked at pages reported below in the MS with tracked changes (PlosOne_Revision2_Edited.doc).

Reviewer #1: The authors have addressed my comments and provided a corrected revision. I have no further objections. Thank you for sharing your interesting findings!

Thank you very much.

Reviewer #2: The authors have thoroughly revised the manuscript, which is now much clearer, more methodologically rigorous, and easier to follow.

I have only one final comment for the authors, which concerns one of the early points in the Discussion section. Specifically, the authors state that "participants of all groups were faster initiating a forward movement... suggesting an alcohol avoidance bias." However, the results show a main effect of stimulus type, but not an interaction with movement direction. Therefore, I do not believe it is appropriate to interpret this as evidence of an avoidance bias. Rather, the data seem to indicate a general slowing in response to alcohol-related stimuli, regardless of movement direction.

Apart from this, in my view the paper is now ready for publication.

We thank Reviewer 2 for this comment. This was updated at page 17.

---

## [Editor Report · Decision Letter 2]

Investigating approach/avoidance tendencies in male AUD patients through a gait initiation task: An exploratory posturography study

PONE-D-24-55094R2

Dear Dr. Campanella,

We’re pleased to inform you that your manuscript has been judged scientifically suitable for publication and will be formally accepted for publication once it meets all outstanding technical requirements.

Kind regards,

Alessandro Mengarelli

Academic Editor

PLOS ONE

Additional Editor Comments (optional):

The additional comments have been properly addressed by the authors. The paper is now suitable for being published.
---

## [Editor Report · Acceptance letter]

PONE-D-24-55094R2

PLOS ONE

Dear Dr. Campanella,

I'm pleased to inform you that your manuscript has been deemed suitable for publication in PLOS ONE. Congratulations! Your manuscript is now being handed over to our production team.

Kind regards,

on behalf of

Dr. Alessandro Mengarelli

Academic Editor

PLOS ONE